# Recent Advances and Progress of Conducting Polymer-Based Hydrogels in Strain Sensor Applications

**DOI:** 10.3390/gels9010012

**Published:** 2022-12-25

**Authors:** Vinh Van Tran, Kyungjun Lee, Thanh Ngoc Nguyen, Daeho Lee

**Affiliations:** 1Department of Mechanical Engineering, Gachon University, Seongnam 13120, Republic of Korea; 2NTT Hi-Tech Institute, Nguyen Tat Thanh University, 300A Nguyen Tat Thanh, Ward 13, District 4, Ho Chi Minh City 700000, Vietnam

**Keywords:** hydrogels, conducting polymers, strain sensors, e-skin devices, stretchable sensors

## Abstract

Conducting polymer-based hydrogels (CPHs) are novel materials that take advantage of both conducting polymers and three-dimensional hydrogels, which endow them with great electrical properties and excellent mechanical features. Therefore, CPHs are considered as one of the most promising platforms for employing wearable and stretchable strain sensors in practical applications. Herein, we provide a critical review of distinct features and preparation technologies and the advancements in CPH-based strain sensors for human motion and health monitoring applications. The fundamentals, working mechanisms, and requirements for the design of CPH-based strain sensors with high performance are also summarized and discussed. Moreover, the recent progress and development strategies for the implementation of CPH-based strain sensors are pointed out and described. It has been surmised that electronic skin (e-skin) sensors are the upward tendency in the development of CPHs for wearable strain sensors and human health monitoring. This review will be important scientific evidence to formulate new approaches for the development of CPH-based strain sensors in the present and in the future.

## 1. Introduction

Smart wearable and stretchable strain sensing devices have been broadly developed for various applications including health monitoring, human motion detection, sensory skin, and soft robotics because they can recognize subtle changes under external stimuli and record them in electrical signals [1,2,3]. Up to now, various materials such as metals or metal oxides, graphene and its derivatives, carbon nanotubes (CNTs), carbon blacks, nanofibers, and nanoparticles have been employed for fabricating flexible sensors [4]. However, these materials lack stretchability, stability, and adhesion, thus highly limiting their utility in strain sensors and human health monitoring [5,6]. In addition, wearable strain sensors also demand a high self-healing ability for extending their lifetime, durability, and biocompatibility for long-time integration on the human skin and in tissues [7]. Hydrogels are three-dimensional polymers with porous structures that can endow them with good flexibility and stretchability [8,9]. Moreover, they also exhibit biocompatibility and good mechanical properties, which are similar to human skin tissues [10]. Therefore, hydrogels have been recently regarded as potential substrates for the design of flexible and stretchable strain sensors [11,12].

Conductive hydrogels are defined as special hydrogels which have a high conductivity due to integration of conductive materials into the hydrogel matrix [13]. A variety of conductive materials such as graphite materials, carbon nanotubes (CNTs), free ions, liquid metals, and conducting polymers (CPs) have been used for the synthesis of conductive hydrogels. Among them, CPs have been regarded as one of the most promising materials for the development of conductive hydrogels due to their good electrical properties and soft mechanical properties [14]. Conducting polymer-based hydrogels (CPHs) has also been demonstrated as a promising platform for the design of wearable and stretchable strain sensors in practical applications. Most previous review papers have focused on general discussions about conductive hydrogels with a wide range of conducting materials [15,16,17,18] or different applications in supercapacitors [19], biomedical electronics [20], adsorbents, and drug delivery [21]. To our knowledge, there has been no published comprehensive review of the studies on CPHs in strain sensing applications. Therefore, this study intends to provide significant scientific evidence as well as emphasize the importance of CPHs in strain sensing sensors and human health monitoring. Here, this review gives a critical review of outstanding properties and synthetic technologies, along with the recent progress of CPH-based strain sensors in human motion monitoring. Moreover, the fundamentals, principles, and working mechanisms are also summarized and discussed deeply. In view of the contributions in scientific research and the industrial development of CPHs, we hope that this study will benefit the discovery of potential approaches for the development and implementation of CPHs in practical strain sensors.

## 2. Properties of CPHs

CPHs have been commonly used in sensing applications because they possess many outstanding properties, which are summarized in Table 1. This section discusses some of the distinction features of CPHs, which render them suitable for wearable and stretchable strain sensors.

### 2.1. Electrical Conductivity

Conventional hydrogels often show an intrinsic ionic conductivity from 10^−5^ to 10^−1^ S/cm in physiological conditions [22], which is 6–9 orders of magnitude lower than the conductivity of metals [23], while, CPHs offer both electronic and ionic conductivity, which renders them as potential materials in electronic sensing devices [24]. The conductivity of CPHs varies from 0.3 to 27 S/m depending on the types of conductive polymers or doped materials [25]. Undoped polymers have low conductivities (10^−6^–10^−10^ S·cm^−1^) at the boundary region between semiconductor sand insulators, while dopped polymers show much higher conductivities (>10^4^ S·cm^−1^) [14]. For improvement of the electrical properties, highly conductive materials such as carbon-based materials (e.g., graphite materials and CNTs ), free ions, conjugated polymers, or liquid metals have often been incorporated into the polymer network [26]. CPHs have highly electric conductivity due to the presence of ionic and/or electronic conductive fillers into the hydrogel matrix. CPHs prepared by doping with metal salts or ionic liquids have the lowest conductivity, while CPHs containing conductive nanomaterials such as carbon- and metal-based materials and MXene exhibit both ionic and electronic conductivity and their conductivity is 1–5 orders of magnitude higher [20]. Due to their high conductivity, CPHs provide a wide strain range, high gauge factor (GF), and good stability for strain sensor devices, which broadly extends their practical application in wearable devices.

### 2.2. Mechanical Properties

Compared to conventional hydrogels with brittleness, CPHs showed good mechanical properties with tunability. The porosity of CPHs can highly influence their mechanical properties because the porosity can affect the free volume content, size, connectivity, and surface properties [27]. CPHs with a hollow nanosphere structure and porous nanostructures can achieve good mechanical properties. As an example, Zhang et al. prepared a highly stretchable porous polyvinyl alcohol (PVA) hydrogel possessing both high tensile and compressive strains, 400% and 80%, respectively (Figure 1a) [28]. The outstanding mechanical properties of the porous CPHs can be ascribed to the novel binary network structure or physically cross-linked networks. In addition, porous CPH enables the excellent mechanical properties to be displayed due to hydrogen bonding interactions and crystallization points [29].

Good mechanical strength is considered as one of the most crucial factors for CPH-based strain sensors as well for maintaining their integrity and capability under various stress conditions. Their mechanical properties have been significantly improved by employing a double network (DN) method. The DN-based CPHs show outstanding mechanical strength and elasticity owing to the special features in their contrasting structural networks, strong interpenetrating network entanglement, and efficient energy dissipation [30]. In the DN approach, two polymers with completely different or opposite physical characteristics form an interconnected network [31]. The first network is a rigid framework and the second one is a ductile substance. Therefore, DN-CPHs are mainly constructed from two interpenetrating cross-linked networks which are connected into a soft material matrix and their mechanical features can be easily tuned by varying the compositions of each network. The DN-CPHs exhibit excellent mechanical properties because of the sacrificial bond of the first network, promoting energy dissipation and providing a large extension to protect the second network under high stress conditions [32]. DN-CPHs can be both tough and soft with high failure tensile stress and strain, hardness, and toughness [31,33,34]. As an illustration, Zhao et al. developed a multifunctional ionic DN-CPH with good stretchability by a combination of oxide sodium alginate (OSA), aminated gelatin (AG), and acrylic acid (AA) (Figure 1b) [35]. The physical metal coordination and dynamic Schiff base bonds strengthen this CPHs’ stretchability.

**Figure 1 gels-09-00012-f001:**
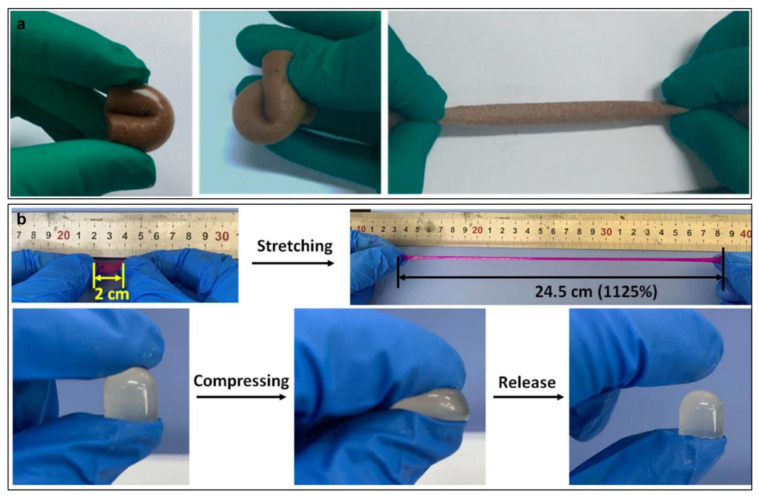
(**a**) Porous CPHs with PVA and GO exhibit extraordinary mechanical properties: bend, curl and stretch (reprinted with permission from ref. [28]); and (**b**) mechanical features of PAA–OSA–AG hydrogels under manual stretching and compressing (reprinted with permission from ref. [35]).

### 2.3. Self-Healing Property

Sensor electronic devices can lose or reduce performance in cases of mechanical damage or under repeated deformation, resulting in the shortening of their service life. Therefore, the self-healing properties are also one of the most important characteristics for CPH-based sensor devices. CPHs with a good self-healing ability can repair their structure, mechanical properties, or other functionalities after damage [36]. Generally, the self-healing ability of CPHs can be attributed by two main types of covalent bonds including dynamic covalent bonds [37,38] and non-covalent bonds. The non-covalent bonds can include various types, i.e., host−guest interaction, metal coordination, a hydrogen bond, and hydrophobic interaction [39,40,41]. In comparison with covalent bonds, the dynamic covalent bonds exhibit both reversibility and stability [42]. Therefore, dynamic covalent bonds including imine bonds [43,44], acylhydrazone bonds [45,46], phenylboronate esters [47,48], disulfide bonds [38,49], and Diels−Alder reactions [46,49] are often preferred for the fabrication of self-healing CPHs. From these, the imine bond from the Schiff base reaction has been widely utilized for the dynamic covalent bond because of its simplicity and the lack of a requirement for catalyst assistance [50]. Moreover, the Schiff base reaction can be processed simply and fast under neutral conditions and without external stimulus. For instance, Lei et al. recently prepared a biocompatible gelatin-based self-healing CPH using the imine bond obtained by the Schiff base reaction between gelatin and cellulose (CMC) without an additional cross-linking agent (Figure 2a) [36]. Due to a large number of amino groups in gelatin, the Schiff base cross-linkages can be promoted to form a hydrogel by the dynamic imine bonds between gelatin and CMC, which resulted in the high flexibility and softness of hydrogels under pressing and bending. Moreover, the hydrogel can self-heal after cutting and rejoining in 30 min and the healed hydrogel can withstand significant tension (Figure 2b). The healing proficiency can reach 90% after 60 min healing. Therefore, CPHs can possess excellent self-healing characteristics.

### 2.4. Adhesion Property

In addition to their excellent mechanical characteristics, CPHs also possess enough adhesion ability which is a crucial feature for sensitive sensing capabilities. Due to strong and stable adhesion on the substrates and devices, the interfacial failures and functionality loss of CPHs can be inhibited, which substantially increases the reliability and efficacy of the sensor devices [51]. Moreover, CPHs can be attached to various solid materials in the design of sensing devices, including metals, ceramics, glass, silicon, polymers, or even human skin [52]. Hydrogels are often hydrophilic, while most flexible substrates are hydrophobic. For achieving the strong adhesion properties of CPHs, two main factors need to be adapted: excellent mechanical properties and strong interfacial bonding between the hydrogel and substrate. Therefore, stealth or inert materials have been commonly used for fabricating self-adhesive CPHs because they can prevent biomacromolecule adhesion and still preserve the hydrogel functions [53]. For example, methacrylic acid (MAA) containing vinyl and carboxyl groups in its structure, has been recently introduced in CPHs for enhancement of hydrogels’ adhesion to various substrate surfaces [54]. It has been indicated that the increased adhesion strength of the CPHs to some substrate surfaces (copper, polytetrafluoroethylene, and skin) can be achieved when increasing the amount of MAA due to the polar groups’ interaction between the amide and carboxylic acid on the hydrogel.

CPHs with great adhesion and biocompatibility can be regarded as promising candidates for developing strain sensors in the human health monitoring. The adhesion of CPHs in these applications is ascribed by carboxyl and catechol functional groups in the hydrogel matrix through the formation of hydrogen bonding, hydrophobic action, metal complexation, and π–π stacking with the hydroxyl, carboxyl, and amino groups on the substrates [55]. Moreover, these physical interactions also make CPHs show highly repeatable adhesion. Recently, Ma’s group reported a novel CPH with strong adhesion by using bases in adherent DNA [56]. This self-healing hydrogel was produced by three chemical cross-linking networks including a G−C base pairing hydrogen bond, chemical cross-linking, and an ionic cross-linking dynamic network. This CPH can be strongly adhered to different surfaces (i.e., metals, glass, plastics, natural materials, and animal and human skin) (Figure 3). In addition, the CPH presented a good skin affinity and maintained great adhesion under the deformation process.

**Figure 3 gels-09-00012-f003:**
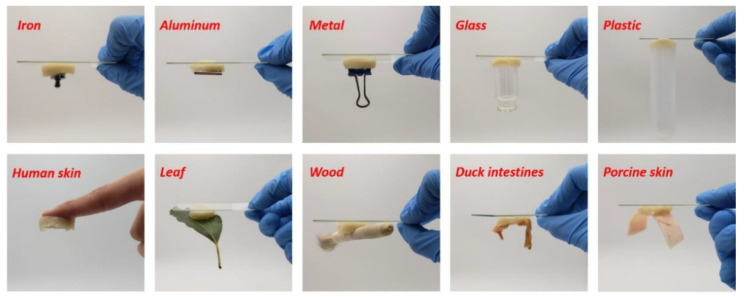
Adhesive property of the CaSA–GAD hydrogel to different surfaces (reprinted with permission from ref. [56]).

**Table 1 gels-09-00012-t001:** Summary and comparison of the outstanding properties of CPHs for strain sensing applications.

Properties of CPHs	General Description	Advantages for Strain Sensors	Refs
Electrical conductivity	- The conductivities of CPHs vary from 0.3 to 27 S/m - High electric conductivities due to the presence of ionic and/or electronic conductive fillers into the hydrogel matrix	Providing wide strain ranges, high gauge factors (GFs), and excellent stability	[20,25,26]
Mechanical properties	- The mechanical properties are affected by the porosity through free volume content, size, connectivity, and surface properties - High tensile and compressive strains, i.e., ~400% and ~80%, respectively	Maintaining the integrity and capability under various stress conditions.	[27,28,29]
Self-healing properties	- The self-healing ability of CPHs can be attributed to two main types of covalent bonds including dynamic covalent bonds and noncovalent bonds - The Schiff base reaction is a well-known covalent bond in CPHs	Repairing the structure, mechanical property, or other functionalities after damage	[37,38,50]
Adhesion properties	- Stealth or inert materials are widely used for the preparation of self-adhesive CPHs - Carboxyl and catechol functional groups are ascribed for the strong adhesion of CPHs	Inhibiting interfacial failures and functionality loss of CPHsIncreasing the reliability and efficacy	[51,52,53,54]

## 3. Preparation Methods of CPHs

### 3.1. Blending or Doping

Direct doping or blending conducting components into hydrogel matrices has been considered as one of the most common and simple techniques for the preparation of CPHs. In this approach, traditional hydrogels are mixed with metal-based particles, carbon nanoparticles, and conducting polymers during the polymerization of conductive polymers [17]. For instance, Zhu et al. successfully fabricated a polyion complex/polyaniline (PIC/PANI) composite CPH using the simple blending method [57]. The CPH was prepared by mixing a conducting phase polymer (PANI) with a PIC matrix and phytic acid (Figure 4a) and the obtained hybrid hydrogels showed the high conductivity and the viscoelasticity of the tough matrix. It has been demonstrated that this facile method can be possible for the large-scale and fast fabrication of CPHs. Nonetheless, the mixing or doping strategy exhibited several disadvantages such as aggregation of the conductive components, reduced mechanical performance, low conductivity, and transparency because of the high specific surface energy of the conducting nanomaterials [58]. Therefore, some active agents or surfactants are often introduced to promote the distribution of the dispersed phase and reduce the aggregation through specific molecular interactions between the conducting agents and polymer matrix [59].

Doping engineering has recently been considered as a simple and effective technique for the synthesis and modulation of CPHs [60]. In this approach, some vital features of CPHs, i.e., electronic features, microstructures, and structural-derived mechanical properties can easily be modulated and tuned by varying dopants or doping levels during the synthesis process. For instance, one-dimensional nanostructured polypyrrole (PPy)-based CPHs were effectively controlled by using a rational dopant counterion, copper phthalocyanine-3,4′,4″,4‴-tetrasulfonic acid tetrasodium salt (CuPcTs) (Figure 4b) [61]. The CuPcTs play a dual role (dopant and gelator) to make self-assembly with the PPy in nanostructured CPHs. Interestingly, using the doping approach, the PPy CPHs enable fabrication on a large scale with the uniform morphology of 1D nanofibers. In addition, various CPH morphologies, i.e., nanoparticles, nanofibers, and foam-like nanostructures, can be tailored by using the different structures of CuPcTs dopant molecules because of the steric effects and electrostatic interaction (Figure 4c). The use of doping engineering can produce smart CPHs with controllable properties, which are expected to have great potential for applications in various sensor technologies [62]. The doping method has been regarded as one of the most vital strategies to tailor and improve the conductivity, mechanical strength, and stability of CPHs [63,64]. Among numerous dopants, polymer dopants have gained more popularity and polymer-doped CPHs are great candidates to develop electronic skin (e-skin)-based applications [65].

**Figure 4 gels-09-00012-f004:**
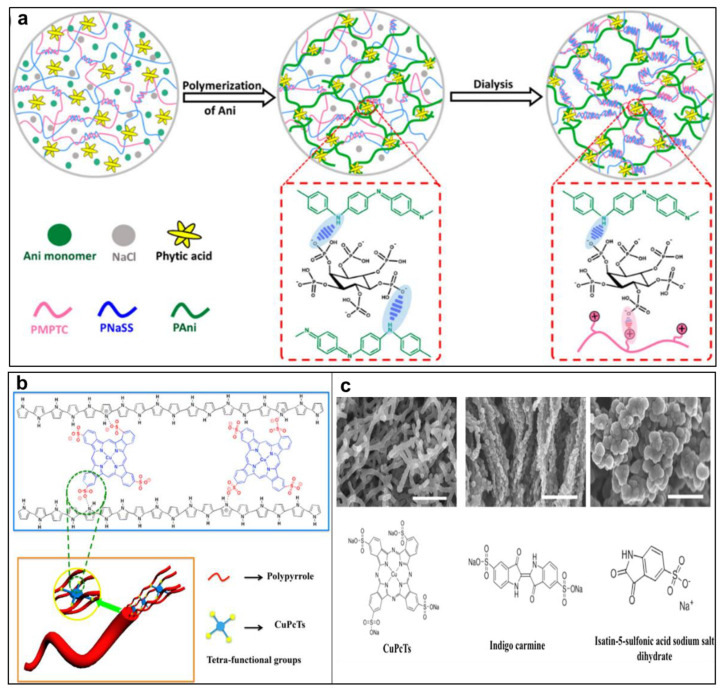
(**a**) The preparation process of the PIC/PANI composite hydrogel by blending of PANI, phytic acid, and NaCl (reprinted with permission from ref. [57]); (**b**) synthesis of nanostructured PPy CPHs using a doping method; and (**c**) SEM images of various nanostructured PPy hydrogels with different doping agents (reprinted with permission from ref. [61]).

### 3.2. Copolymerization

In the case of the polymer-based doping method, CPHs are prepared by polymerization of CP monomers onto the nanostructured insulating hydrogel templates. Alternatively, another approach has been employed for preparation of CPHs by copolymerizing CPs and insulating polymers to form a composite hydrogel. In other words, the copolymerization reaction generates CPHs by taking the unique properties of monomers, polymer chains, or components in the resultant hydrogel composite [66,67]. The copolymerization has been widely employed for fabricating CPHs which possess multifunctional features, i.e., conductivity, stretchability, stability, self-healing properties, and biocompatibility [68,69]. Among various copolymerization approaches, direct copolymerization is of great significance in the synthesis of CPHs due to its simplicity and efficiency. In this technique, monomers are simply mixed at a specific ratio and the reaction is activated by adding gelators or changing the external conditions [70]. This approach can offer a facile methodology to produce CPHs with the desired properties by controlling the content of monomers during the copolymerization process. For instance, Jiang et al. developed a mechanical-tunable CPH by copolymerizing functionalized poly (ethylene dioxythiophene) (f-PEDOT) with two vinyl monomers, acrylic acid (AA) and hydroxyethyl methacrylate (HEMA) (Figure 5) [71]. The results demonstrated that the physical properties of this CPH, i.e., swelling ratios and mechanical characteristics, enable it to be easily tailored by modifying the chemical composition of the hydrogel network during the polymerizing process. Compared with other advanced techniques, however, copolymerization also showed several disadvantages such as low mechanical strength [72] and a lack of long-term stability and practical applicability [73].

### 3.3. Advanced Techniques for the Preparation of CPHs

The development of novel synthetic technologies has been one of the most important strategies in the improvement of CPHs’ practical applications. It is still a huge challenge for the preparation of CPHs which possess the desired architectures and satisfy the requirements of real applications. Three-dimensional printing technology has recently gained great attention and is considered as an emerging versatile additive manufacturing technology in the design of CPHs due to its advantages of rapid prototyping and customizability [74,75]. Compared with traditional techniques, 3D printing technology exhibits some superior advantages: a facile operation procedure, precise structural control, and cost-effectiveness [76]. In this technology, CPHs with 3D structures and various functionalities enable them to be formed ingeniously and individually by a layer-by-layer printing process [77,78]. Three-dimensional printing technologies are generally classified into five main types based on their design and working mechanism, including: stereolithography (SL), digital light processing (DLP), bioplotting, inkjet printing, and optical printing method [79,80]. Among them, SL and DLP are two methods that show a high spatial resolution, commercialization, and practical applications [81]. Nonetheless, SL has two drawbacks: long-time processing and high cost. In contrast, DLP technology shows short-time, low-cost, high-accuracy, and multi-material processing [15,82], thus it has been employed as an outstanding model of 3D printing technology in the fabrication of CPHs. For instance, Caprioli et al. used a commercial DPL printer to prepare a 3D CPH with an excellent self-healing ability [83]. In another study, Wei et al. also designed a bioinspired CPH by incorporating CNTs into a hydrogel matrix of polyacrylic acid (PAA) and sodium alginate (SA) using DLP printing technology (Figure 6) [84]. Due to the 3D printing method, this bioinspired CPH showed excellent stretchability and multiple conductivities and was easily integrated in a strain sensor with simultaneous piezoresistive and piezocapacitive performances. In addition, the CPH can be explored as an advanced hydrogel in multifunctional skin-like smart wearable devices [84]. 

## 4. Design Principles and Working Mechanism of Wearable Flexible CPH-Based Strain Sensors

CPHs have been proven to be a great candidate for strain sensor applications because they can sensitively respond to external stimuli by changing their electrical properties and network structures. Specifically, the conductive network of CPHs will be denser or sparser as a pressure or strain is applied, respectively, resulting in a decrease or increase in resistance or conductivity (Figure 7a) [16,85]. Moreover, a synergistic effect between the hydrogel matrix and conductive networks plays an important role in the sensing performance of strain sensors. Sensitivity is a critical parameter for evaluating the performance of CPH strain sensor devices, which highly depends on the network structures. The limit of changed conductivity and gauge factor (GF) are two main factors used to measure the sensitivity of strain sensors. As mentioned above, the former is induced by the density or integrity, while the latter comes from the breaking or deformation of the CPH network. Therefore, a wonderful balance of these two factors should be deeply considered when designing a CPH strain sensor. Moreover, repeatability is also another important parameter which presents the stable response of strain sensors. 

These days, numerous CPHs have been applied to design flexible and wearable strain sensors for monitoring human activities and health. A wide range of human physiological signals, i.e., heartbeat, breath, expression changes, finger touch, peristole, vocal-cord vibration, and joint bending are able to be recognized and recorded timely and accurately using CPH-based strain sensors [86,87,88,89]. These CPH strain sensors precisely transform small strains from joints, the epidermis, and cardiac tissue into electronic signals. They are also wearable and stretchable sensors attached to the human skin to measure the movement signals of different body parts in real time. For human movement and health monitoring applications, there are different working mechanisms which can be assigned for the excellent sensing performance of wearable CPH-based strain sensors, such as piezoresistivity, disconnection, the tunneling phenomenon, crack propagation, and piezo capacitance [3,90,91,92]. CPHs prepared by composites of CPs with semiconductors, CNTs and metal oxides often work on the piezoresistivity mechanism [85]. The disconnection mechanism is known as separating overlapped, rigid conductive fillers in the CPH matrix, while the tunneling effect occurs when the closely spaced conductive fillers (nanographene, CNT, and Ag nanowire) are disentangled in the CPH network [93,94]. Finally, crack propagation happens as a layer of conductive fillers in the hydrogel network is damaged and recovered under stretching [91]. For enhancing the working range and sensitivity, CPH-based strain sensors are often designed based on the integration of several sensing mechanisms [94].

The piezoresistivity mechanism is considered as one of the most important working mechanisms in CPH-based strain sensors. Figure 7b presents a detailed illustration of the piezoresistivity of a PVA–PPy CPH strain sensor [91]. When applying a small strain, the deformation of the PVA network results in a local crack in the conductive PPy paths, increasing the resistance. However, the released stress will recover these cracks as well as resistance. In the piezoresistivity mechanism, the conductor content plays an important role in the sensing performance. A low or excessive number of conductors (PPy) in the hydrogel matrix will cause the easy destruction or blockage of the conductive paths, leading to reduced change in resistance and poor sensitivity under small strain. The sufficient content of conductors makes the conductive paths of the CPH network reversibly crack and recover under the stretching/releasing process, resulting in high sensitivity and good repeatability.

**Figure 7 gels-09-00012-f007:**
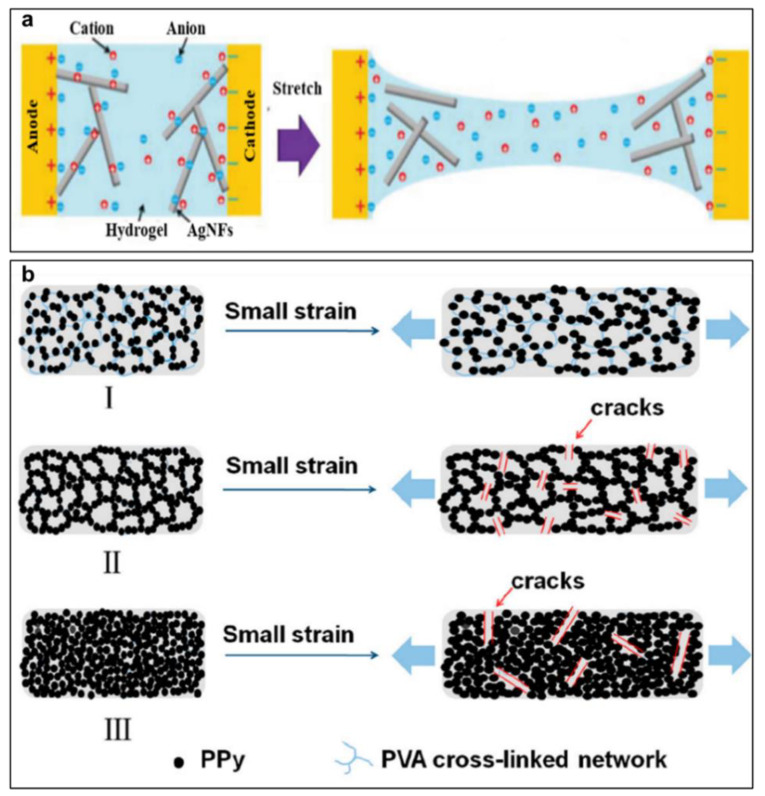
(**a**) Schematical illustration of a strain sensor based on CPHs (reprinted with permission from ref. [16]); and (**b**) the piezoresistivity mechanism of a PVA–PPy strain sensor when applying small strain (reprinted with permission from ref. [91]).

## 5. Recent Advancements of CPH-Based Strain Sensors

### 5.1. MXenes-CPH-Based Strain Sensors

MXenes are known as the thriving family of transition metal carbides/carbonitrides which possess hydrophilic surfaces and high electrical conductivity [95,96]. They have been recently regarded as potential materials for the design of electromechanical strain sensors with a high sensing capacity owing to their extraordinary mechanical features as well as the largely changed interlayer distance under external stimuli [97,98,99]. Moreover, due to their high specific surface area and abundance in surface functional groups such as–OH, –F, and –O, MXenes nanosheets can be uniformly distributed in the hydrogel network by formation of negatively charged surfaces [100]. Furthermore, these hydrophilic surfaces considerably enhance the mechanical characteristics of CPHs (i.e., elastic modulus and stretchability) due to the formation of a typical clay–polymer network structure [101]. In addition, the large number of hydrogen bonds in the hydrogel matrix are introduced by the high density of surface functional groups, leading to a significant improvement in the self-healing ability of hydrogels [102]. Compared with other commonly used materials in the design of CPHs such as metal- or carbon-based nanoparticles, MXenes show superior advantages owing to the combination of metal conductivity, a larger specific surface area, and excellent stability [103,104,105,106,107]. Therefore, MXenes have been recently employed as revolutionary materials for enhancement of the mechanical properties and conductivities of CPHs [95]. MXenes are often used as conductive fillers in hydrogels. 

MXenes have been combined with a wide range of polymers to prepare CPHs for the development of wearable strain sensors. For instance, Li et al. prepared a multifunctional epidermal sensor by combining the MXene (Ti_3_C_2_T_x_) with a PAA polymer [108]. Due to the presence of MXenes, this CPH sensor exhibited highly stretchable, self-healing, and degradable abilities, as well as a sensitive and fast response to human motions. Luan’s group also demonstrated that the combination of MXenes with a PAM/SA double-network hydrogel produced an CPH-based strain sensor with good tensile properties (2000%) and excellent electrical stability [109]. In another study, Zhang et al. developed an MXene CPH with extremely high stretchability and instantaneous self-healability by integrating it with a PVA network [110]. Moreover, this hydrogel also showed excellent conformability and adhesion to human skin. The results confirmed that the addition of MXene nanosheets considerably increased the strain sensitivity of the CPH. In addition, without a complex circuit design and unique capabilities, the MXene/PVA hydrogel was employed as the sensing film for advanced sensing applications (Figure 8), including: signature recognition, individual handwriting, and vocal sensing. These results indicated that MXene-CPHs will be promising materials for the development of wearable electronics, point-of-care testing, and soft robotics in human motion and health monitoring.

### 5.2. CPH Strain Sensor for Electronic Skin (E-Skin) Devices

Artificial e-skin refers to flexible, stretchable, and self-healing electronics simulating the functionalities of human skin to perceive subtle stimulus signals [111]. E-skins with multimodality, high flexibility, and biocompatibility have recently attracted great attention in the development of soft robotics, prosthetics, and human−machine interfaces [112,113,114]. For precise stimulus recognition and the detection of multi-signals, multimodal sensing platforms have been integrated into stretchable and transparent e-skin devices [115,116]. Moreover, this is necessary for the development of wearable, attachable, or implantable e-skin devices effectively integrating with the human skin for practical applications [117]. In particular, e-skin strain sensors need to sensitively convert external stimuli to electric signals for identification of the precise position where they are attached [118].

Owing to outstanding transparency (99% visible-light transmission) and minimal variation in their conductive properties under minor strain–stress loops, CPHs have been considered as a potential material for the design of e-skin devices [119,120]. Such CPH e-skin devices are constructed by a flexible electronic device containing a hydrogel film and two or three electrodes, and they are attached to human skin [120]. Moreover, CPHs also have stretchability, biocompatibility, a self-healing ability, and long-term stability, which renders them completely suitable for the fabrication of smart devices that can be transplanted onto the human body [121,122]. Therefore, CPHs have started to be used for the development of e-skin microsensors with multi-functions (e.g., good adhesion, high elasticity, good compatibility, fast response, and human skin-like protective performance) in a diversity of fields such as artificial intelligence, human health detection, and soft robotics [123,124,125]. Recently, CPHs have been demonstrated as an ideal platform for e-skin sensors with good stretchability like human skin and strong adhesion on human tissue, which reduces interface resistance and motion artifacts. Moreover, skin irritation, rubefaction, and pain problems during the use of CPH e-skin devices were also avoided. For instance, Liu and co-workers successfully prepared a skin-biocompatible CPH containing large hydrogen bonds by combining PVA, phytic acid (PA), and gelatin (Gel) [126]. The PVA/PA/Gel (PPG) composite CPH hydrogel exhibited good adhesion, easy detachability, and good adaptability during wrist stretching and bending (Figure 9a,b). The PPG hydrogel also showed better features than commercial hydrogels (Figure 9c). In addition, the PPG hydrogel e-skin sensor was regarded as having valuable application prospects to recognize electrophysiological signals in human motion and health monitoring due to its high transparency, breathability, antimicrobial activity, low cost, and recyclability.

## 6. Conclusions

CPHs possess many superior features in both electrical and mechanical properties over conventional hydrogels as well as other materials, which qualifies them as a potential platform for the design of wearable and stretchable strain sensors in human health monitoring. They have both electronic and ionic conductivity. Moreover, CPHs exhibit high mechanical strength and stretchability, which can also be tunable in a simple manner. The self-healing property is also one of the most crucial and interesting characteristics of CPHs, which allows them to repair their structure and functionalities. Finally, their excellent adhesion ability and biocompatibility are distinct features as well that have increasingly gained attention among the general public and the scientific community in the design of strain sensor devices for human motion detection and health monitoring. For the preparation technologies of CPHs, doping or blending and copolymerizing methods are common and simple techniques used to synthesize composite-based CPHs or improve the conductivity of CPHs based on conventional hydrogels. On the other hand, the technique of 3D printing is considered as an advanced technology to produce CPHs with novel structures and remarkable features. The use of MXenes and the design of e-skin devices for human heath monitoring are promising development strategies for the fabrication and application of CPHs in strain sensors. However, there have been several challenges and there are additional future directions for the development of CPHs in advanced strain sensor devices: (i) the development of advanced CPHs that possess all the following characteristics, i.e., biocompatibility, strong mechanical properties, anti-freezing properties, strong adhesion, a self-powered ability, ultrafast self-healing and recovery, and transparence; and (ii) CPH-based sensors need to be highly responsive and correctly distinguish tiny human motions such as heart beats for employment in practical applications in human health care.

## Figures and Tables

**Figure 2 gels-09-00012-f002:**
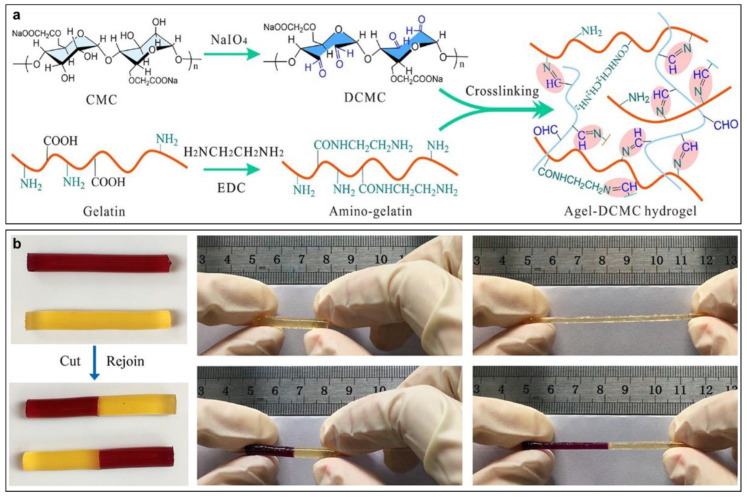
(**a**) Fabrication route of self-healing gelatin−cellulose (CMC) hydrogel based on the Schiff base reaction; (**b**) a cut hydrogel self-heals after rejoining; and the stretching ability of the healed hydrogel (reprinted with permission from ref. [36]).

**Figure 5 gels-09-00012-f005:**
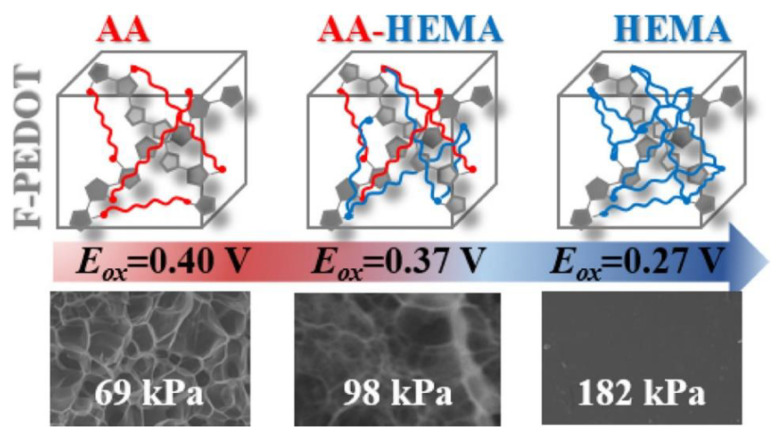
Fabrication process of conducting hydrogels using a copolymerization approach with vinyl monomers (reprinted with permission from ref. [71]).

**Figure 6 gels-09-00012-f006:**
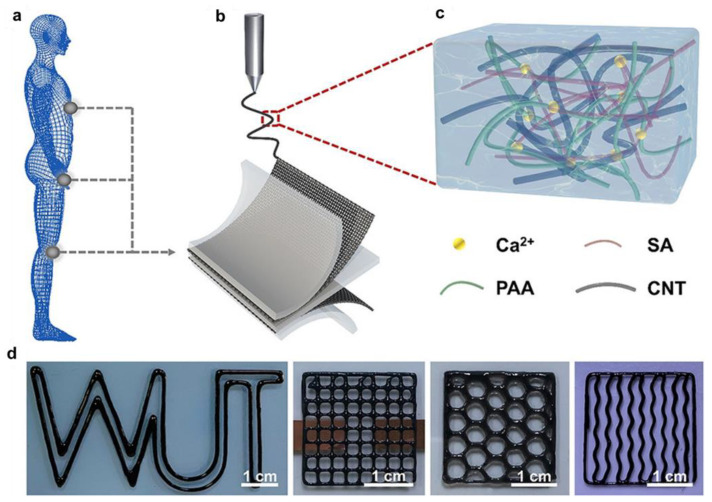
The process was used to prepare the Ca–PAA-=–SA–CNTs hydrogel-based strain sensor: (**a**) Detecting human body movements; (**b**) 3D printed process for preparation of the Ca-PAA-SA-CNTs hydrogels and skin-like strain sensor; (**c**) Ca-PAA-SA-CNTs hydrogels; (**d**) 3D printing of Ca-PAA-SA-CNTs hydrogels in various patterns. (reprinted with permission from ref. [84]).

**Figure 8 gels-09-00012-f008:**
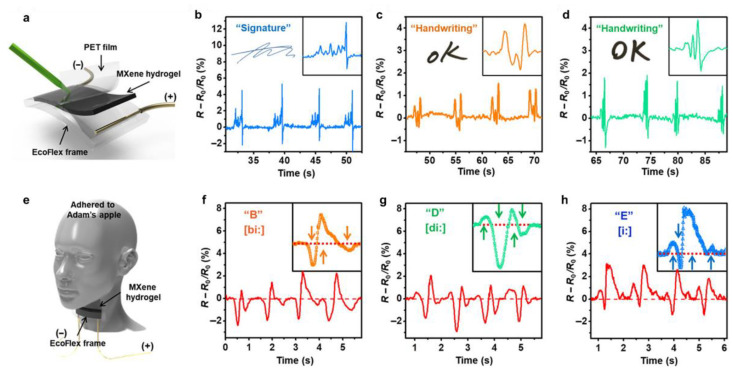
Advancement in strain sensors based on MXenes/PVA hydrogel: (**a**) schematical illustration of signature sensing; changed resistance to (**b**) signature; and (**c**,**d**) handwriting. (**e**) Schematical illustration of vocal sensing. (**f**–**h**) Changed resistance in responding to similar sounding letters (reprinted with permission from ref. [105]).

**Figure 9 gels-09-00012-f009:**
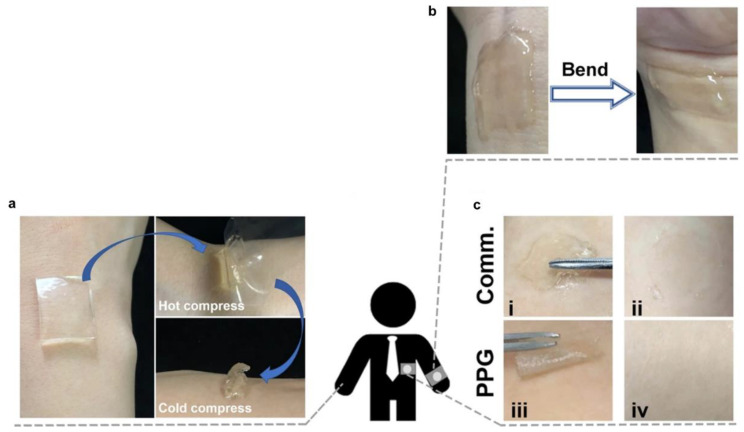
PPG hydrogel-based strain e-skin sensor: (**a**) photographs of PPG CPH adhering to and peeling off of the skin; (**b**) adaptability of PPG CPH to human skin; and (**c**) photographs of a commercial hydrogel product (**i**,**ii**) and PPG CPH (**iii**,**iv**) which are peeled off from the human skin (reprinted with permission from ref. [126]).

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
