# Peer review of "Recent Advances and Progress of Conducting Polymer-Based Hydrogels in Strain Sensor Applications"

_gels, 2022, doi:10.3390/gels9010012_

Round 1

Reviewer 1 Report

The article can be accepted after minor revision

1. The authors need to explain the advantages of conductive polymers over other materials in the introduction part

2. Electrical conductivity conductive polymers need to be addressees with published proof data means values

3. Elasticity owing to the special features in their contrasting structural networks- if elasticity owing property influences the  conductivity

4. How Self-healing property affects the sensor activity need to be explained

5. Adhesion property how it is crucial property need to be explain

6.  What is the difference Copolymerization and electropolymerization?

7. Advanced techniques for preparation of CPHs- reference must be include new techniques

8. What about the repeatability of the sensors

9. CPHs possess many superior features- these features discussed in table with comparison

10.  If possible please include sensitivity table

11.  What are the advantages of the review article regarding sensor developments?

Author Response

We would like to thank all reviewers for investing their time in reviewing the manuscript and for providing insightful comments that we believe have significantly improved our manuscript. Please find below our responses to the specific comments made by the reviewers. The changes made to the manuscript are highlighted in YELLOW.

Reviewer #1

The article can be accepted after minor revision.

  1. The authors need to explain the advantages of conductive polymers over other materials in the introduction part

 Response: We highly appreciate your suggestion. In the introduction section, the advantages of conductive polymers was emphasized and compared with other conducting materials such as graphite materials, carbon nanotubes (CNTs), free ions, liquid metals.

- Page 1, line 39-45: “A variety of conductive materials such as graphite materials, carbon nanotubes (CNTs), free ions, liquid metals, and conducting polymers (CPs) have been used for synthesis of conductive hydrogels. Among them, CPs have been regarded as one of the most promising materials for development of conductive hydrogels due to their good electrical properties and soft mechanical properties [14]. Conducting polymer-based hydrogels (CPHs) has also been demonstrated as a promising platform for design of wearable and stretchable strain sensors in practical applications”

  1. Electrical conductivity of conductive polymers need to be addressees with published proof data means values

 Response: We highly appreciate your suggestion. We added the conductivity of undoped and dopped polymers along with the references in the manuscript.

- Page 2, line 73-75: “Undoped polymers have low conductivities (10−6–10−10 S·cm−1) at the boundary region between semiconductors and insulators, while dopped polymers show much higher conductivities (> 104 S·cm−1) [14]”

  1. Elasticity owing to the special features in their contrasting structural networks- if elasticity owing property influences the conductivity

 Response: We highly appreciate reviewer’s comment. The double network (DN) CPHs show outstanding mechanical strength and elasticity owing to the special features in their contrasting structural networks, strong interpenetrating network entanglement, and efficient energy dissipation [24]. The DN CPHs are composed of two contrasting polymer network structures. The first network is a highly crosslinked polyelectrolyte as a rigid scaffold to maintain their shapes, and the second network is loosely crosslinked flexible neutral polymers as the filler of the rigid scaffold to absorb external stresses. Hence, DN hydrogels exhibit soft and rigid characteristics as well as high conductivity. In other words. The elasticity and conductivity of DN CPHs are completely independent.

  1. How Self-healing property affects the sensor activity need to be explained

 Response: We highly appreciate reviewer’s comment. We note that the self-healing property is also one of the most important characteristics for CPHs-based sensor devices. CPHs with good self-healing ability can repair their structure, mechanical property, or other functionalities after damage. Therefore, self-healing property do not directly affect the sensor activity, however, it contributes to extend the service life of the sensor devices.

  1. Adhesion property how it is crucial property need to be explained

Response: We highly appreciate reviewer’s comment. We emphasized importance of the adhesion property in strain sensing applications.

-Page 4, line 154-157: “CPHs also possess an enough adhesion ability which is a crucial feature for sensitive sensing capabilities. Due to strong and stable adhesion on substrates and devices, interfacial failures and functionality loss of CPHs can be inhibited, which substantially increases the reliability and efficacy of sensor devices [45]. Moreover, CPHs can be attached to various solid materials in the design of sensing devices, including metals, ceramics, glass, silicon, polymers, or even human skin [46]”.

  1. What is the difference Copolymerization and electropolymerization?

 Response: We highly appreciate reviewer’s comment. Copolymerization and electropolymerization are widely used techniques for synthesis of polymers. Electropolymerization is initiated by the oxidation of a monomer in an electrochemical cell, followed by the growth of the polymer. Electropolymerization is an advantageous and simple method used to synthesize conducting polymers. Copolymerization is often used to prepare a new polymer with properties different from the constituent homopolymers. Copolymerization are widely used to fabricate double network (DN) CPHs for strain sensors. The electropolymerization is one of  the most common approaches of the copolymerization method in term of an electrochemical copolymerization

  1. Advanced techniques for preparation of CPHs- reference must be include new techniques

 Response: We highly appreciate reviewer’s suggestion. In the section “3.3. Advanced techniques for preparation of CPHs”, we discussed 3D printing technologies including 5 main types: stereolithography (SL), digital light processing (DLP), bioplotting, inkjet printing, and optical printing methods. These technologies are the sate-of-the -art techniques for preparation of CPHs integrated in strain sensor devices.

  1. What about the repeatability of the sensors

 Response: We highly appreciate reviewer’s suggestion. Repeatability is also an important parameter of strain sensors. Thus we added it in the manuscript

- Page 9, line 303-304: “Moreover, repeatability is also another important parameter which presents a stable response of strain sensors”

  1. CPHs possess many superior features- these features discussed in table with comparison

 Response: We highly appreciate reviewer’s suggestion. According to your view, we added a table to summarize and compare outstanding features of CPHs in strain sensing applications.

- Page 5, Table 1 as below:

Table 1. Summary and comparison of outstanding properties of CPHs for strain sensing applications

Properties of CPHs

General description

Advantages for strain sensors

Refs

Electrical conductivity

- The conductivities of CPHs vary from 0.3 to 27 S/m

- High electric conductivities due to the presence of ionic and/or electronic conductive fillers into the hydrogel matrix

Providing wide strain ranges, high gauge factors (GFs), and excellent stability

[20,25,26]

Mechanical properties

- The mechanical properties are affected by the porosity through free volume content, size, connectivity, and surface properties

- High tensile and compressive strains, i.e., ~ 400% and ~80%, respectively

Maintaining the integrity and capability under various stress conditions.

[27-29]

Self-healing property

- The self-healing ability of CPHs can be attributed by two main types of covalent bonds including dynamic covalent bonds and noncovalent bonds

- Schiff base reaction is a well-known covalent bond in CPHs

Repairing the structure, mechanical property, or other functionalities after damage

[37,38,50]

Adhesion property

- Stealth or inert materials are widely used for preparation of self-adhesive CPHs

- Carboxyl and catechol functional groups are ascribed for the strong adhesion of CPHs

- Inhibiting interfacial failures and functionality loss of CPHs

- Increasing the reliability and efficacy

[51-54]

  1. If possible, please include sensitivity table

 Response: We highly appreciate reviewer’s comment. This review aims to provide a critical overview of distinct features, preparation techniques, and recent advancements of CPH-based strain sensors. We do not give a generally literature summary about recent applications of CPHs-based strain sensors. Therefore, we suppose that a sensitivity table is not necessary in this present study.

  1. What are the advantages of the review article regarding sensor developments?

Response: We highly appreciate reviewer’s comment. As mentioned above, the present study provides a critical review of distinct features and preparation techniques, and advancements of CPH-based strain sensors for human motion and health monitoring applications. Moreover, fundamentals, working mechanisms, and requirements for design of CPH-based strain sensors with high performance are also summarized and deeply discussed. Therefore, this review will offer new approaches as well as important scientific evidences for development of CPH-based strain sensors in the present and future.

Reviewer 2 Report

The authors discuss the fundamentals, working mechanism, and design of strain sensors based on conductive polymer-based hydrogels (CPHs). The manuscript is an interesting and meaningful overview of the application of conductive polymer-based hydrogels. It can certainly be useful to Gel readers to expand their horizons in the field of practical application of conductive polymer-based hydrogels. I have a few suggestions for the authors to make their manuscript even more meaningful and useful to the scientific community. Thus, the reviewer suggests publication on the Gels after minor revisions.

1. General Comments: Language and style need serious revision. Informal phrases and repeated sentences, such as "Thank you...". Especially the authors should explain their subjective ideas in detail, please correct it.

2. Abstract - Please provide a comprehensive abstract that covers the problem and the study's objective, as well as materials and methods, results, and conclusions. The abstract written in this form acts as an exhaustive statement of results without qualitative explanation and partially explain the significance of the paper, and properly address the accomplished conclusion. Please reshape it.

3. Introduction - Authors should be given about the previous related studies done. Also, provide detailed and informative information about published articles. At the end provide the importance of the study and objectives selected for the study.

4. The chemical structures of the polymers and the responsive mechanisms should be included.

5. Result and discussion part - In this section authors should provide detailed information regarding finding and outcome of research with numerical data and correlate it with previously published studies. The introductory sentences need to be avoided. Please reshape it.

6. Conclusion should be enriched with an outlook devoted to challenges facing CPHs and future development directions.

Author Response

Detailed Response to Reviewer’s comments

We would like to thank all reviewers for investing their time in reviewing the manuscript and for providing insightful comments that we believe have significantly improved our manuscript. Please find below our responses to the specific comments made by the reviewers. The changes made to the manuscript are highlighted in YELLOW.

 Reviewer #2

The authors discuss the fundamentals, working mechanism, and design of strain sensors based on conductive polymer-based hydrogels (CPHs). The manuscript is an interesting and meaningful overview of the application of conductive polymer-based hydrogels. It can certainly be useful to Gel readers to expand their horizons in the field of practical application of conductive polymer-based hydrogels. I have a few suggestions for the authors to make their manuscript even more meaningful and useful to the scientific community. Thus, the reviewer suggests publication on the Gels after minor revisions.

  1. General Comments: Language and style need serious revision. Informal phrases and repeated sentences, such as "Thank you...". Especially the authors should explain their subjective ideas in detail, please correct it.

 Response: We highly appreciate reviewer’s comment. It has been invited a native English-speaker to proofread the revised manuscript by correction of the mistakes and grammatical errors carefully, as your suggestion. Moreover, we carefully checked and corrected all the subjective ideas in the throughout manuscript. 

  1. Abstract - Please provide a comprehensive abstract that covers the problem and the study's objective, as well as materials and methods, results, and conclusions. The abstract written in this form acts as an exhaustive statement of results without qualitative explanation and partially explain the significance of the paper, and properly address the accomplished conclusion. Please reshape it.

 Response: We highly appreciate reviewer’s comments. We revised the abstract in a good shape according to your suggestions, as below.

- Page 1, Abstract: “Conducting polymer-based hydrogels (CPHs) are novel materials that take advantage of both conducting polymers and three-dimensional hydrogels, which endow them in great electrical properties and excellent mechanical features. Therefore, CPHs are considered as one of the most promising platforms for employing wearable and stretchable strain sensors in practical applications. Herein, we provide a critical review of distinct features and preparation technologies, and advancements of CPH-based strain sensors for human motion and health monitoring applications. Fundamentals, working mechanisms, and requirements for design of CPH-based strain sensors with high performance are also summarized and discussed. Moreover, recent progress and development strategies for implementations of CPH-based strain sensors is pointed out and described. It has been supposed that electronic skin (e-skin) sensors are the upward tendency in development of CPHs for wearable strain sensors and human health monitoring. This review will be important scientific evidence to formulate new approaches for development of CPH-based strain sensors in the present and future.”

  1. Introduction - Authors should be given about the previous related studies done. Also, provide detailed and informative information about published articles. At the end provide the importance of the study and objectives selected for the study.

 Response: We highly appreciate reviewer’s comment. The introduction was rewritten and revised according to your suggestion in term of summarizing the published review as well as clarifying the importance and objectives of this study.

- Page 2, line 49-61: “Most previous review papers have focused on general discussion about conductive hydrogels with a wide range of conducting materials [15-18] or different applications in supercapacitors [19], biomedical electronics [20], adsorbents, and drug delivery [21]. To our knowledge, there has been no published comprehensive review of the studies on CPHs in strain sensing applications. Therefore, this study intends to provide significantly scientific evidence as well as emphasize importance of CPHs in strain sensing sensors and human health monitoring. Here, this review gives a critical review of outstanding properties and synthetic technologies, along with the recent progress of CPH-based strain sensors in human motion monitoring. Moreover, fundamentals, principles, and working mechanisms are also summarized and deeply discussed. In view of contributions in scientific research and industrial development of CPHs, we hope that this study will benefit to discovery potential approaches for the development and implementation of CPHs in practical strain sensors.”

  1. The chemical structures of the polymers and the responsive mechanisms should be included.

 Response: We highly appreciate reviewer’s comment. Chemical structures of conducting polymers have been mentioned and discussed in some published review papers (Nanoscale, 2015, 7, 12796-12806; Polymers 2018, 10, 1078). Therefore, we think that adding chemical structures of conducting polymers in this study is not necessary. In term of responsive mechanisms, we have explained it in the manuscript in the section “4. Design principles and working mechanism of wearable flexible CPHs-based strain sensors”, as below:

- Page 9, line 300 – 326: “These CPH strain sensors precisely transform small strains from joints, the epidermis, and cardiac tissue into electronic signals. They are also wearable and stretchable sensors attached to the human skin to measure the movement signals of different body parts in real time. For human movement and health monitoring applications, there are different working mechanisms which can be assigned for excellent sensing performance of wearable CPH-based strain sensors, such as piezoresistivity, disconnection, tunneling phenomenon, crack propagation, and piezocapacitance [3,87-89]. The piezoresistivity mechanism is considered as one of the most important working mechanisms in CPHs-based strain sensors. Figure 7b presents a detailed illustration of the piezoresistivity of a PVA-PPy CPH strain sensor [88]. When applying a small strain, the deformation of PVA network results in a local crack of the conductive PPy paths, increasing the resistance. But the released stress will recover these cracks as well as resistance. In the piezoresistivity mechanism, the conductor content plays an important role in the sensing performance”

  1. Result and discussion part - In this section authors should provide detailed information regarding finding and outcome of research with numerical data and correlate it with previously published studies. The introductory sentences need to be avoided. Please reshape it.

 Response: We highly appreciate reviewer’s comment. We note that this study is a review paper, which provide a critical review of distinct features and preparation technologies, and advancements of CPH-based strain sensors. Therefore, finding and outcome of research with numerical data and correlate it with previously published studies have been not presented. However, we carefully checked and corrected the introductory or general sentences in the throughout manuscript according to your comment.

  1. Conclusion should be enriched with an outlook devoted to challenges facing CPHs and future development directions.

 Response: We highly appreciate reviewer’s suggestion. The conclusion was enriched and revised by adding some challenges and future directions of CPHs according to your comment, as below:

- Page 13: “CPHs possess many superior features in both electric and mechanical properties over conventional hydrogels as well as other materials, which qualifies them as a potential platform for design of wearable and stretchable strain sensors in human health monitoring. They have both electronic and ionic conductivity. Moreover, CPHs exhibit high mechanical strength and stretchability, which can also be tunable in simple manners. Self-healing property is also one of the most crucial and interesting characteristics of CPHs, which allows them to repair their structure and functionalities. Finally, excellent adhesion ability and biocompatibility are as well distinct features that has increasingly gained attention among the general public and the scientific community in the design of strain sensor devices for human motion detection and health monitoring. For preparation technologies of CPHs, doping or blending, and copolymerizing methods are common and simple techniques used to synthesize composites based CPHs or improve the conductivity of CPHs based on conventional hydrogels. On the other hand, the 3D printing technique is considered as an advanced technology to produce CPHs with novel structures and re-markable features. The use of MXenes and design of e-skin devices for human heath monitoring are promising development strategies for fabrication and application of CPHs in strain sensors. However, there has been several challenges as well as future directions for development of CPHs in advanced strain sensor devices: (i) development of advanced CPHs that possess all the following characteristics, i.e., biocompatibility, strong mechanical properties, anti-freezing property, strong adhesion, self-powered ability, ultrafast self-healing and recovery, and transparence. (ii) CPH-based sensors need to highly response and correctly distinguish tiny human motions such as heart beats for employing practical applications in human health care.”

Reviewer 3 Report

Reviewer report on Manuscript Draft ‘Recent Advances and Progress of Conducting Polymer-based Hydrogels in Strain Sensor Applications’

The present study provides a critical review of distinct features and preparation techniques, and advancements of Conducting polymer-based hydrogels based strain sensors for human motion and health monitoring applications. Moreover, fundamentals, working mechanisms, and requirements for design of Conducting polymer-based hydrogels based strain sensors with high performance are also summarized and discussed. It is predicted that electronic skin (e-skin) sensors are the upward tendency in development of Conducting polymer-based hydrogels for wearable strain sensor and human health monitoring. It was also concluded that the use of MXenes and design of e-skin devices for human heath monitoring are promising development strategies for fabrication and application of Conducting polymer-based hydrogels in strain sensors.

This manuscript is in the scope of journal is interestingly addressed and well written. Therefore, the manuscript eventually can be published after some improvements:

Recent reviews on conducting polymers  could be overviewed and discussed in Introduction part of the manuscript.

Authors are concluding that the use of MXenes and design of e-skin devices for human heath monitoring are promising development strategies for fabrication and application of Conducting polymer-based hydrogels in strain sensors. Therefore some reviews on application of MXenes (Progress and Insights in the Application of MXenes as New 2D Nano-Materials Suitable for Biosensors and Biofuel Cell Design. International Journal of Molecular Sciences 2020, 21, 9224.) could be overviewed and discussed.

Author Response

Detailed Response to Reviewer’s comments

We would like to thank all reviewers for investing their time in reviewing the manuscript and for providing insightful comments that we believe have significantly improved our manuscript. Please find below our responses to the specific comments made by the reviewers. The changes made to the manuscript are highlighted in YELLOW.

Reviewer #3

Reviewer report on Manuscript Draft ‘Recent Advances and Progress of Conducting Polymer-based Hydrogels in Strain Sensor Applications’

The present study provides a critical review of distinct features and preparation techniques, and advancements of Conducting polymer-based hydrogels-based strain sensors for human motion and health monitoring applications. Moreover, fundamentals, working mechanisms, and requirements for designing polymer-based hydrogels based strain sensors with high performance are also summarized and discussed. It is predicted that electronic skin (e-skin) sensors are the upward tendency in development of Conducting polymer-based hydrogels for wearable strain sensor and human health monitoring. It was also concluded that the use of MXenes and design of e-skin devices for human heath monitoring are promising development strategies for fabrication and application of Conducting polymer-based hydrogels in strain sensors.

This manuscript is in the scope of journal is interestingly addressed and well written. Therefore, the manuscript eventually can be published after some improvements:

  1. Recent reviews on conducting polymers could be overviewed and discussed in Introduction part of the manuscript.

 Response: We highly appreciate reviewer’s comment. Recently published reviews of researches on conducting polymers and conducting hydrogels were summarized and discussed in the Introduction section as below.

- Page 2, line 49-61: “Most previous review papers have focused on general discussion about conductive hydrogels with a wide range of conducting materials [15-18] or different applications in supercapacitors [19], biomedical electronics [20], adsorbents, and drug delivery [21]. To our knowledge, there has been no published comprehensive review of the studies on CPHs in strain sensing applications. Therefore, this study intends to provide significantly scientific evidence as well as emphasize importance of CPHs in strain sensing sensors and human health monitoring. Here, this review gives a critical review of outstanding properties and synthetic technologies, along with the recent progress of CPH-based strain sensors in human motion monitoring. Moreover, fundamentals, principles, and working mechanisms are also summarized and deeply discussed. In view of contributions in scientific research and industrial development of CPHs, we hope that this study will benefit to discovery potential approaches for the development and implementation of CPHs in practical strain sensors.”

  1. Authors are concluding that the use of MXenes and design of e-skin devices for human heath monitoring are promising development strategies for fabrication and application of Conducting polymer-based hydrogels in strain sensors. Therefore some reviews on application of MXenes (Progress and Insights in the Application of MXenes as New 2D Nano-Materials Suitable for Biosensors and Biofuel Cell Design. International Journal of Molecular Sciences 2020, 21, 9224.) could be overviewed and discussed.

Response: We highly appreciate reviewer’s suggestion. MXenes are novel 2D materials that has been considered as promising materials for development of CPHs in strain sensors We inserted and discussed the suggested study as ref [99] in the manuscript, at Page 11, line 346.
